# Effect of Juglone against *Pseudomonas syringae* *pv Actinidiae* Planktonic Growth and Biofilm Formation

**DOI:** 10.3390/molecules26247580

**Published:** 2021-12-14

**Authors:** Qiqi Han, Luoluo Feng, Yani Zhang, Runguang Zhang, Guoliang Wang, Youlin Zhang

**Affiliations:** College of Food Engineering and Nutrition Science, Shaanxi Normal University, Xi’an 710119, China; hqq@snnu.edu.cn (Q.H.); fengluoluo@snnu.edu.cn (L.F.); zyn15771911933@163.com (Y.Z.); sunshine@snnu.edu.cn (R.Z.)

**Keywords:** *Pseudomonas syringae pv Actinidiae*, juglone, reactive oxide species (ROS), extracellular polymers (EPS), biofilm

## Abstract

*Pseudomonas syringae**pv Actinidiae* (*P. syringae*) is a common pathogen causing plant diseases. Limoli proved that its strong pathogenicity is closely related to biofilm state. As a natural bacteriostatic agent with broad-spectrum bactericidal properties, juglone can be used as a substitute for synthetic bacteriostatic agents. To explore the antibacterial mechanism, this study was carried out to examine the inhibitory effect of juglone on cell membrane destruction, abnormal oxidative stress, DNA insertion and biofilm prevention of *P. syringae*. Results showed that juglone at 20 μg/mL can act against planktogenic *P. syringae* (10^7^ CFU/mL). Specially, the application of juglone significantly damaged the permeability and integrity of the cell membrane of *P. syringae*. Additionally, juglone caused abnormal intracellular oxidative stress, and also embedded in genomic DNA, which affected the normal function of the DNA of *P. syringae*. In addition, environmental scanning electron microscope (ESEM) and other methods showed that juglone effectively restricted the production of extracellular polymers, and then affected the formation of the cell membrane. This study provided a possibility for the development and utilization of natural juglone in plants, especially *P. syringae*.

## 1. Introduction

Bacterial crop diseases seriously affect agricultural production all over the world [1,2]. Around the world, there are about 500 species of pathogens inducing crop diseases. As the important plant pathogen, *P. syringae* has 41 varieties and a wide host range, including a variety of woody and herbaceous plants, causing leaf spots, necrosis, and stem ulcers [3,4].

As an aerobic Gram-negative pathogen, *Pseudomonas syringae pv Actinidiae (**P. syringae**)* can easily cause an actinidia canker [5,6]. This disease will increase the fatality rate of actinidia trees, and seriously affect the yield and quality of actinidia [7,8]. At first, the common copper-containing bactericides and streptomycin were effective in treating the disease. However, drug-resistant strains occurred over time, therefore the bactericidal effectiveness of these traditional fungicides has been gradually diluted [7,9,10]. The occurrence of drug resistance is mainly due to the effect of the extracellular polymer (ESP) of *P. syringae*. Extracellular polymers are substances that surround bacteria and provide structural support, adhesion, and information transfer functions for cell membranes. Due to the emergence of drug-resistant strains, it is becoming more and more difficult to control *P. syringae* [4]. Therefore, it is necessary to develop effective intervention measures for control.

Plant-based natural bacteriostatic agents can effectively treat drug-resistant strains and have the characteristics of low toxicity. Juglone is a kind of naphthoquinone substance extracted from walnut green husk, and has broad-spectrum antibacterial properties, such as anti-bacterial, insecticidal, anti-inflammation, anti-cancer, anti-virus, hypoglycaemic, and therapeutic effects on human skin diseases [11,12,13]. Additionally, juglone can induce the apoptosis of pathogens by inhibiting protease synthesis, disrupting redox metabolism, and destroying nucleic acid [12]. As the waste after nut processing, juglone has the characteristics of environmental safety, low cost, and obvious effect. Additionally, the antibacterial performance of juglone remains stable with time, and the possibility of producing drug-resistant strains is low [11]. Therefore, juglone has broad application prospects to apply strong bacteriostatic properties to the treatment of plant diseases.

This work explored the antibacterial mechanism of juglone against *P. syringae* from the aspects of cell membrane destruction, nucleic acid lysis damage and inhibition of extracellular polymer formation. Therefore, this study will present an effective method for the treatment of kiwifruit ulcer disease, and also provide new ideas and new materials for the treatment of other plant bacterial infections.

## 2. Results

### 2.1. Antibacterial Activity of Juglone against P. syringae

As shown in Table 1, the values of DIZ, MIC, and MBC of the *P. syringae* treated with juglone was 22.3 ± 1.00 mm, 20 μg/mL, 50 μg/mL, respectively. The addition of juglone at 1/2MIC significantly and inhibitory affected the growth of *P. syringae* (Figure 1). Compared to the control, there was no obvious colony growth of *P. syringae* in the first 2 h, as the concentration of juglone increased to MIC and MBC (*p* < 0.01).

### 2.2. Effect of Juglone on the Vitality of P. syringae

The entire field was bright blue with no red dots, indicating thenormal cell morphology and structure of *P. syringae* in the control. As the concentration of juglone increased to 1/2MIC and MIC, the blue bright spots gradually decreased and the red bright spots gradually increased, indicating that part of the cell structure of *P. syringae* was damaged. When the concentration of juglone increased to MBC, the entire field was almost entirely red, indicating that all cells of *P. syringae* no longer had normal structure and function (Figure 2).

### 2.3. Effect of Juglone on the Cell Morphology of P. syringae

In the control, the cells of *P. syringae* had a clear rod-shaped structure with saturated cells, a smooth surface, and no adhesion between cells (Figure 3). In the 1/2MIC group, the cells of *P. syringae* began to deform, with surface shrinkage and fracture, and some contents began to extravasate. When the concentration of juglone reached MIC, the cell shape of *P. syringae* was completely deformed, with the surface completely collapsed and fragmented, a large number of contents leaked out, and intercellular adhesion occurred. Cell membrane structure of *P. syringae* was completely lost in the MBC treatment group.

### 2.4. Juglong Induced Membrane Destruction in P. syringae Cells

#### 2.4.1. Effect of Juglone Induction on Cell Membrane Potential of *P. syringae*

The difference in ion concentrations on both sides of the cell membrane causes the cell membrane potential (MP) change, which affects the metabolic activity of the cell. In this study, the variation in bacteria MP was represented by the change in fluorescence intensity of rhodamine 123. As shown in Figure 4A, the MP of the juglone treatment group was significantly lower than that of the control group (*p* < 0.01). Compared with the fluorescence intensity of the control group, whose fluorescence intensity of 1/2MIC, MIC and MBC groups decreased by 24.17%, 67.5% and 69.27%, respectively, which indicated that juglone could effectively destroy the membrane potential.

#### 2.4.2. Effects of Juglone Induction on Intracellular Protein and Nucleic Acid Contents of *P. syringae*

Protein and nucleic acid are the basic components of all living things and play an important role in the life activities of organisms. As can be seen from Figure 4B, extracellular protein content in the treatment group increased significantly and reached the peak at about 8 h. Compared with the control group, the difference was extremely significant in the juglone treatment group (*p* < 0.01). The change trend of intracellular protein content was contrary to the above mentioned. In Figure 4F, the intracellular DNA concentrations of *P. syringae* were 65.8, 49.2, and 19.2 μg/mL after additions of juglone at concentrations from 1/2 MIC to MBC. The OD260 nm values increased from 0.21 to 0.34 to 0.73 and 0.96 after juglone treatment (Figure 4E). There were significant differences between the administration groups and the control group (*p* < 0.01). The results showed that the treatment of juglone can cause the obvious leakage of intracellular protein and nucleic acid, which will lead to the destruction of intracellular structure, and then cause the disorder of cell function.

#### 2.4.3. Effect of Juginone Induction on Intracellular ATP Content of *P. syringae*

We further evaluated the effect of juglone on the intracellular ATP content of *P. syringae*. ATP is an important substance that provides energy for the metabolism of various physiological activities in the cell and the transportation of substances across the membrane [14]. As shown in Figure 4D, the intracellular ATP decreased by 41.47%, 65.55% and 88.96% in juglone treatment groups, respectively. Additionally, the difference was significant compared with the control (*p* < 0.01). When the concentration of juglone reached MBC, the intracellular ATP content was as low as 26.954 μmol/g, which was too low to support any live activities of bacteria. 

### 2.5. Intracellular Oxidative Stress Caused by Juglone

Fluorescence intensity of DCFH-DA was measured by inverted fluorescence microscope and fluorescence spectrophotometer. As shown in Figure 5, the control group maintained a low level of ROS content throughout the process, while the treatment group continued to increase the fluorescence intensity of DCFH-DA in the lilac leaf cells in a dose-dependent manner. As the treatment time reached 6 h, the fluorescence intensity of MIC and MBC groups reached 89,000 and 120,000, respectively, which were about 4 times and 6 times of the control group, and the difference was extremely significant (*p* < 0.01).

### 2.6. Effect of Juglone on DNA Structure

#### 2.6.1. Fluorescence Spectra Analysis of DNA

The fluorescence intensity in the 610 nm control group was significantly higher than that in the other treatment groups, and the fluorescence intensity gradually decreased with the increase in the concentration of juglone (Figure 6A). The reason may be that small molecules of juglone competitively insert into the base pair of DNA, causing the base pair to fall off and the fluorescence intensity to decrease [15]. The DNA gel electrophoresis (Figure 6C) showed that the intensity of the bands in the drug treatment group was significantly shallow compared with that in the control group, which suggested that the nucleic acid content was reduced. This is consistent with the conclusion that fluorescence intensity decreased, which confirmed that juglone can effectively reduce the content of DNA.

#### 2.6.2. CD Spectra Analysis

CD spectroscopy is an effective technique to study the structural changes of DNA. It can be clearly seen from Figure 6B that the DNA of *P. syringae* had a negative band at 240 nm and a positive band at 270 nm. The negative band at 240 nm was due to the dextral helicity of DNA, and the positive band at 270 nm is due to the base accumulation, which is a DNA-B conformational structure characteristic. The intensity of positive and negative bands decreased obviously with the increase in the concentration of juglone, and the negative bands shifted slightly. DNA gel electrophoresis also showed that the DNA in the administration group was broken up into small fragments, indicating that the secondary structure of *P. syringae* was destroyed (Figure 6C).

### 2.7. Effect of Juglone on P. syringae Biofilm Formation

#### 2.7.1. Determination of Fluorescence Intensity of Crystal Violet

As shown in Figure 7, *P. syringae* has the ability to form biofilm on the surface of the test surfaces. Biofilm formation in each treatment group decreased by 25.61%, 37.8%, 62.2% and 68.29%, respectively, compared with that of the control group. The juglone 1/4MIC treatment group significantly inhibited the formation of biofilm (*p* < 0.01), and the formation of *P. syringae* biofilm decreased with the increase in juglone concentration.

#### 2.7.2. Effect of Juglone Induction on Extracellular Polymer Content of *P. syringae* and ESEM Observation of Biofilm Formation

We further detected changes in protein, polysaccharide mucus and DNA content in EPS (Figure 7). In general, the contents of protein, polysaccharide slime and DNA in EPS decreased with the increase in juglone concentration, in a dose-dependent manner. Among them, the protein content was only 8.65 μg/mL when the concentration of juglone increased to MIC, which decreased more than 8 times compared with the control group. The difference was extremely significant (*p* < 0.01). The content of polysaccharide mucus changed most obviously with the concentration of juglone. With the concentration of juglone increasing to 1/2MIC, polysaccharide slime content was only 16.7% of the control group. When the concentration of juglone was increased to MIC, the content of polysaccharide slime was only 12.2 μg/mL, significantly reduced by 93% compared with the control group. Similar changes were seen in DNA content. When the concentration of juglone was MIC, the difference was extremely significant compared with the control group. At the same time, ESEM images also obtained similar results, the cell morphology and structure of the untreated group were normal, the biofilm was dense, and the cells were wrapped in EPS structure. At the concentration of 1/2MIC, the biofilm structure was destroyed, and the extracellular polymer disappeared. With the concentration of juglone increased to MIC and MBC, the cell biofilm structure was completely destroyed.

## 3. Discussion

The formation of *P. syringa**e* biofilms and the emergence of drug-resistant strains have seriously threatened the yield and quality of crops [16]. There is an urgent need to find new bacteriocin pairs against *P. syringa*. As an effective combat tool, juglone can effectively inhibit the growth of pathogenic bacteria and the formation of biofilm. In this context, we studied the inhibitory effect of juglone on *P. syringae* [17,18]. The results showed that juglone had an irreversible and inhibitory effect on the growth of *P. syringae* (10^7^ CFU/mL) (Figure 1). Especially, the bacterial membrane of *P. syringae* was completely destroyed with the leakage of a large amount of intracellular material, and finally the cell structure was lost (Figure 3). Therefore, juglone can act as an effective combat tool to inhibit the growth of pathogenic bacteria.

As an important natural barrier, bacterial biofilm of cells provides a stable and safe environment for their own growth and also resist external infections. MP plays an important role in bacterial physiology. Especially, a decrease in MP of bacteria revealed the depolarization of cell membrane, which induced the reduction of the volume of bacteria [7]. In addition, as an element of the proton motive force, MP is involved in the generation of ATP [14]. Therefore, the membrane potential was chosen as an aspect to illustrate the mechanism of antibacterial action.

Biological macromolecules play an important role in cells, and any loss of macromolecules causes irreversible inhibition on cells [19]. The destruction of membrane potential leads to functional damage or even rupture of the cell membrane, which leads to the leakage of intracellular proteins, nucleic acids, ATP and other substances (Figure 4). Another reason for the decrease in intracellular macromolecules may be that juglone inhibits the expression of related genes and blocks the expression of related enzymes and the production of related substances [14,20]. In conclusion, juglone directly destroyed the cell membrane of *P. syringae*, and inhibited the expression of related genes, reducing the levels of intracellular protein, ATP and DNA of *P. syringae*.

ROS such as free radical species and singlet oxygen, is a by-product of the respiring metabolism for normal organisms. Even at low levels, ROS could attenuate the injury of cells, and at high levels, ROS was lethal for cells, even inducing triggers and effector molecules of cell apoptosis [21]. There was more ROS in the bacterial cells with the increase in juglone concentration (Figure 5). When the high dose ROS surpassed the threshold of the anti-oxidative capacity of *P. syringae*, ROS could destruct the bacterial structures and intracellular mediated functions by degrading and deforming the cellular biomacromolecules [22]. Meanwhile, ROS induced the destruction of the structure of nucleic acid, the denaturation of protein and the senescence and death of the cell (Figure 5) [21,23]. These results indicated that juglone induced growth inhibition and death by enhancing oxidative stress in *P. syringae*.

Does juglone affect the nucleic acid itself? We already know that juglone disrupts the bacterial cell membrane leading to a massive release of nucleic acid. From the fluorescence spectra (Figure 6A), we can see that the fluorescence intensity of each juglone treatment group decreased significantly. This suggests that the DNA structure has been attacked by juglone, possibly disrupting the nucleotide sequence. The CD spectrum (Figure 6B) also proved that the DNA structure and helix structure of the DNA of *P. syringae* were destroyed to some extent. It can be seen from Figure 6B that the intensity of both positive and negative bands decreased and both of them were offset. The insertion of juglone into DNA disturbs the helical structure and base accumulation of DNA, thus changing the cadmium spectrum. This will seriously affect the normal expression of genes, block the synthesis of some enzymes in cells, and lead to cell dysfunction and even cell death [16]. At the same time, DNA was obviously cut into small fragments in the MBC treated group in DNA gel electrophoresis (Figure 6C). In summary, as a small molecule, juglone was embedded in DNA structure, which made the helicity of DNA more compact, and the secondary structure was destroyed.

Crystal violet quantitative analysis confirmed that juglone also had a significant blocking effect on the formation of cell membrane (Figure 7). Additionally, ESEM images also clearly showed a large number of cell aggregation and extracellular polymer of *P. syringae* in the control, but in the juglone treatment, cells of *P. syringae* began to disperse and the extracellular polymer decreased (Figure 7A). This result is consistent with other studies [24]. The reasons for the strong diffusion and drug resistance of *P. syringae* are the good adhesion of cell membrane, the diffusion of ESP and the difficulty of antibiotics to penetrate the biofilm (this may be related to the role of EPS as diffusion barrier, molecular sieve and adsorbent) [25].

*P. syringae* biofilms are structured complexes embedded in extracellular polymeric substances (EPS), which can improve the survival rate of microorganisms in extreme environments and also enhance bacterial infectivity [4]. The results showed that juglone can effectively inhibit the accumulation of polysaccharide mucus, protein, and DNA, and then disrupted the production of ESP (Figure 7C). Especially, it has obvious influence on the content of polysaccharide slime. According to relevant reports, extracellular polysaccharide slime provides shape and structure support for biofilm, and also contributes to the adhesion and protection of bacteria [24]. Therefore, the inhibition of polysaccharide slime production by juglone may play an important role in the inhibition of biofilm formation.

## 4. Materails and Methods

### 4.1. Materials and Culture

*Pseudomonas syringae pv Actinidiae* was obtained from the Food Safety and Health Laboratory, Food Engineering and Nutrition Science College, Shaanxi Normal University. Prior the experiment, the activated *P. syringae* strain was cultured in King’s B broth medium (Qingdao Hope Bio-Technology Co., Ltd., Qingdao, China) at 26 °C for 14 h in shaking flask and diluted with King’s B broth until the OD value was approximately 0.5 (approximately 10^7^ CFU/mL). All other reagents used in the experiment were analytically pure. Juglone was extracted from walnut green husk by our previous experimental method, and the extracted juglone was further purified and identified, and the purity was 70.5% (walnut husk in Xi ‘an Roseque market) [26].

### 4.2. Antibacterial Activity Assay of Juglone against P. syringae

#### 4.2.1. Diameter of the Inhibitory Zone Assay

Experimental operations are similar to those described in the previous report by Liu et al. [15]. The bacterial suspension (approximately 10^7^ CFU/mL) was evenly coated on the plate, and 200 μL of 70 µg/mL juglone (the concentration of dissolved DMSO was 0.5% (*w*/*v*)) was injected into the Oxford cup. After incubation at 24 °C for 24 h, DIZ was measured [27].

#### 4.2.2. Minimum Inhibitory Concentration and Minimum Bactericidal Concentration Assays

Specific tests were carried out according to previous methods [28]. The juglone was dissolved in the King’s B broth to a final concentration of 0–60 μg/mL. Then, 50 µL of the bacterial suspension (approximately 107 CFU/mL) was inoculated into each tube and incubated at 24 °C for 24 h. MIC was considered to have no visible bacterial growth in King’s B broth, and MBC was considered to have no colony growth in plate.

#### 4.2.3. Growth Curve Assay

To measure it according to our previous method, in short: juglone was dissolved in LB broth, resulting in the final concentration of juglone at 0, 1/2MIC, MIC and MBC [26]. Immediately, *P. syringae* was inoculated in each tube and cultured at 37 °C. Bacterial growth was monitored every 2 h at 600 nm by spectrophotometer (1530, Thermo Fisher Scientific Oy, Wuhan, China).

### 4.3. Antibacterial Mechanism of Juglone against P. syringae

#### 4.3.1. Confocal Laser Scanning Microscope (CLSM) Assay

The cell viability measurements were consistent with those previously reported, but with some modifications [29]. Different concentrations of juglone (0 (control), 1/4MIC, 1/2MIC and MIC) were mixed with *P. syringae* (approximately 10^7^ CFU/mL) and cultured at 37 °C for 4 h. The precipitated cells were obtained by centrifugation of 1 mL bacterial suspension. The cells were re-suspended with PBS, and 20 μL SYTO9 and PI (propidium iodide) were added, respectively. The cells reacted in the dark for 15 min. Finally, the bacterial suspension was analyzed with an inverted Olympus FV1200 laser confocal scanning microscope (Olympus Corporation, Tokyo, Japan) with an objective lens of ×40.

#### 4.3.2. Field Emission Scanning Electron Microscope (FESEM) Observation and Analysis

Juglone (0 (control), 1/2MIC, MIC and MBC) and bacterial suspension (approximately 10^7^ CFU/mL) were added into LB broth and incubated for 6 h. The 4 mL mixture was centrifuged and washed to obtain cells. The sample was re-suspended with 2.5% glutaraldehyde and placed at 4 °C for 4 h. The samples were eluted successively with 30–100% ethanol for 10 min each time. Next, the samples were mixed with tert-butanol and freeze-dried [30]. Finally, the sample was sprayed with gold and observed under FESEM (SU8220, Hitachi, Tokyo, Japan).

#### 4.3.3. Determination of Cell Membrane Potential (MP)

The effect of juglone on the metabolic activity of bacteria was reflected by changes of bacteria MP. In this study, the variation in bacteria MP can be represented by the change in fluorescence intensity of rhodamine 123. For specific test operation, refer to the report of Comas et al. [31]. The *P. syringae* suspension (approximately 10^7^ CFU/mL) was prepared and treated with juglone at the 0 (control), 1/2MIC, MIC and MBC levels for 4 h. The collected cells and Rhodamine were mixed and left in the dark for 30 min. The sample was washed with PBS, and the fluorescence intensity of sample was measured when at the excitation wavelength was 480 nm and the emission wavelength was 530 nm.

#### 4.3.4. Determination of Intracellular and Extracellular Protein Content

Refer to the experimental method of Rhayour et al. [32]*. P. syringae* suspension (approximately 10^7^ CFU/mL) was prepared and treated with juglone at the 0 (control), 1/2MIC, MIC and MBC levels. A series of total mixture of 1 mL (obtained at 0, 4, 8 and 12 h, respectively.) was centrifuged to get the supernatant. The obtained supernatant was measured with a protein detection kit (Beyotime Biotechnology, Nanjing, China). The sediment was washed by PBS, and the bacterial protein extraction kit (Solarbio, Beijing, China) was used to process the sediment to obtain bacterial intracellular proteins. Similarly, protein content in the extracted precipitate was determined with a protein detection kit.

#### 4.3.5. Effects of Juglone on Intracellular and Extracellular DNA of *P. syringae*

The concentration of nucleic acid in and outside the cell of bacteria was determined by the method reported by Duan et al. [33]. *P. syringae* suspension (approximately 10^7^ CFU/mL) was prepared and treated with juglone at the 0, 1/2MIC, MIC and MBC levels for 0.5 h. The 1 mL sample was centrifuged at 8000 rpm for 5 min and measured the absorbance of the supernatant at OD260. At the same time, DNA concentration in the precipitate was extracted and determined by DNA separation kit (Sangon, Shanghai, China).

#### 4.3.6. Determination of Intracellular ATP Concentration

*P. syringae* suspension (approximately 10^7^ CFU/mL) was prepared and treated with juglone at the 0 (control), 1/2MIC, MIC and MBC levels for 2 h. The 1 mL of sample was centrifuged at 7000 rpm for 5 min. Bacterial residue was re-suspended with PBS and adjusted to approximately 10^6^ CFU/mL. The ATP concentration of the treated *P. syringae* was measured with an ATP detection kit (Jiancheng, Nanjing, China).

### 4.4. Reactive Oxygen Species (ROS) Assay

Intracellular ROS content was detected according to specific steps of DCFH-DA fluorescence kit (Jiancheng, Nanjing, China). Juglone (final concentration 0 (control), MIC and MBC) was added to the preprepared 4 mL of bacterial suspension (approximately 10^7^ CFU/mL) and co-incubated for 6 h. 1 mL samples were taken every 2 h and centrifuged at 7000 RPM for 8 min. The cells sediment was collected, re-suspended in PBS and added DCFH-DA (Jiancheng, Nanjing, China) in a volume ratio of 49: 1 and incubated at 37 °C in darkness for 30 min. The treated samples were observed under an inverted fluorescence microscope and their fluorescence intensity was measured with a fluorescence spectrophotometer (excitation at 502 nm and emission at 530 nm) [34].

### 4.5. Antibacterial Mechanism on P. syringae Genomic DNA

#### 4.5.1. Extraction of Genomic DNA of *P. syringae* and DNA Gel Electrophoresis

The operation before the experiment is the same as 2.3.6 treatment. DNA of *P. syringae* was extracted by bacterial genome extraction kit. The obtained DNA was diluted to a specific concentration with Tris-HCl (0.1 mol/L, pH 8.0). 5 μL of extracted DNA was mixed with 1 μL of buffer. The obtained mixtures were subjected to electrophoresis in agarose gel for 15–20 min.

#### 4.5.2. Circular Dichroic (CD) Spectrum Assay

The extracted DNA was diluted with Tris-HCl (0.1 mol/L, pH 8.0) to 300 μL. Then, a circular dichroic spectrometer (wavelength 210–320 nm) was performed under nitrogen atmosphere (Chirascan, Applied Photophysics Ltd., Leatherhead, Surrey, UK).

#### 4.5.3. Fluorescence Spectra Assay

The mixture of 200 μL DNA and 15 μL ethidium bromide (Zhonghui, Shaanxi, China) was incubated at 37 °C for 25–40 min in the darkness. Then, the fluorescence spectra were recorded using a fluorescence spectrophotometer at room temperature using 535 nm excitation and 550–750 nm scanning wavelength.

### 4.6. Inhibition of Biofilm Formation by Juglone

#### 4.6.1. Biofilm Formation and Quantitative Crystal Violet Assay

Different concentrations of juglone (0 (control), 1/4MIC, 1/2MIC, MIC, MBC) were added to the 10 mL bacterial suspension. The mixed samples were then added to each well of the 96-well polystyrene plate and incubated at 27 °C for 24 h. When the biofilm formed, the sterile PBS was added to each well to remove the planktonic cells, and methanol was added for 15 min to fix the biofilm. The fixed biofilms were stained with 1% crystal violet solution for 10 min (The biofilm can combine with crystal violet solution, and finally the content of biofilm can be inferred from the fluorescence intensity of crystal violet.). Finally, 95% ethanol (200 μL) was added to each well, and absorbance was measured at 570 nm using a Multiskan Spectrum (Multiskan Go; Thermo, Waltham, MA, USA). The measured values were imported into the following formula to calculate the percentage inhibition of biofilm formation: $$\mathrm{Antibiofilm}\;\mathrm{activity}\;\left(\%\right)=\frac{\left(A-B\right)}{\left(A\right)}\times 100$$

In the formula, *A* is absorbance of control; *B* absorbance of test

#### 4.6.2. ESEM Assay

*P. syringae* suspension (approximately 107 CFU/mL) was incubated in 10 mL King’s B broth medium with juglone (0 (control), 1/2 MIC, MIC, MBC) to form biofilms on glass coverslips for 24 h at 26 °C. Then, the sample was further processed by referring to Liu et al. ’s method (2021). Finally, the dehydrated sample was sprayed with gold and observed on ESEM (FEI-Quanta 200, Hillsboro, OR, USA).

#### 4.6.3. Determination of Main Extracellular Polymers Components

Referring to the method reported by Vazquez-Armenta et al. [35] for specific experimental operations, but with some modifications, different concentrations of juglone (0 (control), 1/4MIC, 1/2MIC, MIC) and suspension of 10^7^ CFU/mL *P.*
*syringae* were mixed into a 6-well plate and incubated at 27 °C for 48 h. Biofilm was formed. After the biofilm was formed, the plankton cells were cleaned with PBS. After natural drying, 1 mL PBS was added, the biofilm was destroyed by ultrasonic dispersion method for 15 min, and the supernatant was obtained by 6000 rpm for 30 min. Finally, the extracellular protein content was determined using a protein detection kit (Beyotime Biotechnology, China). The extracellular polysaccharides in biofilms were determined by ANthrond-H2SO4 reagent. DNA extraction kit (Sangon, Shanghai, China) was used to extract DNA, and the absorbance value was measured at OD260 nm.

### 4.7. Statistical Analysis

All assays were performed in triplicate, and the data was expressed as means ± standard deviation and analyzed using IBM SPSS software (version 22.0; SPSS, Inc., Chicago, IL, USA). The difference was calculated using one-way analysis of variance and considered statistically significant at *p* < 0.05.

## 5. Conclusions

This study demonstrated that juglone extracted from walnut husk had obvious antibacterial activity against *P. syringae* and could effectively treat actinidia canker. Juglone destructed the cell membrane potential and caused irreversible damage to the cell membrane, which led to a massive leakage of cytoplasm (protein, nucleic acid, and ATP). In addition, juglone induced the accumulation of intracellular ROS, which led to abnormal intracellular oxidative stress. Juglone can also be inserted into DNA, causing gene expression disorders, and affecting normal cell function. In general, juglone, as a representative natural antimicrobial agent, which also shows great potential in the treatment of other plant bacterial infections.

## Figures and Tables

**Figure 1 molecules-26-07580-f001:**
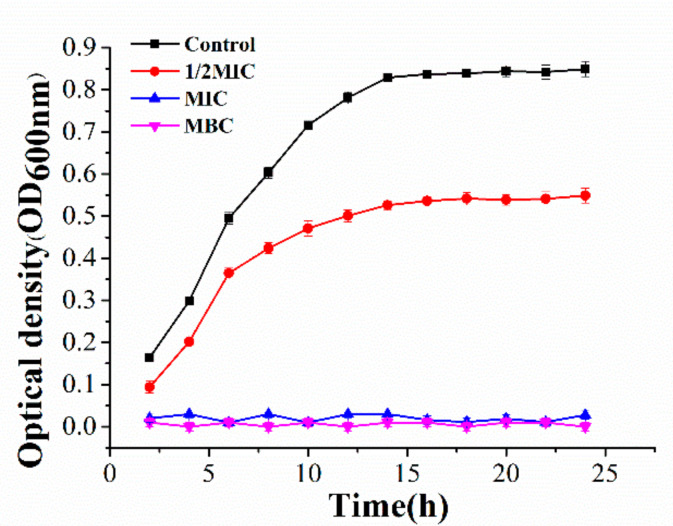
Growth curve of *P. syringae* exposed to juglone.

**Figure 2 molecules-26-07580-f002:**
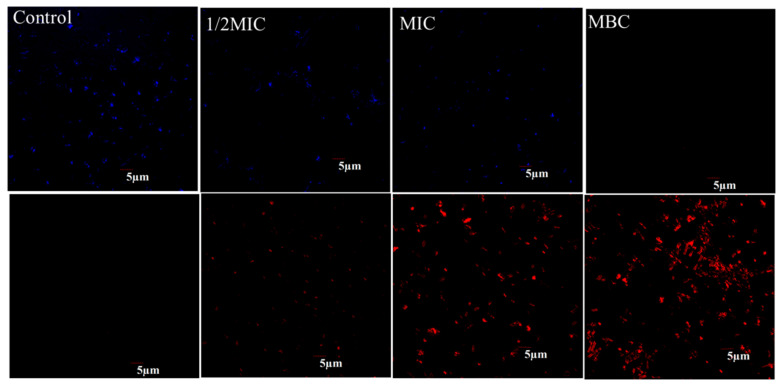
CLSM was used to analyze the viability of *P. syringae* exposed to different concentrations of juglone.

**Figure 3 molecules-26-07580-f003:**
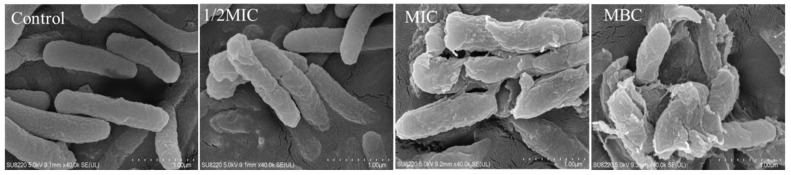
FESEM images (×40,000) of the cell membrane of *P. syringae*.

**Figure 4 molecules-26-07580-f004:**
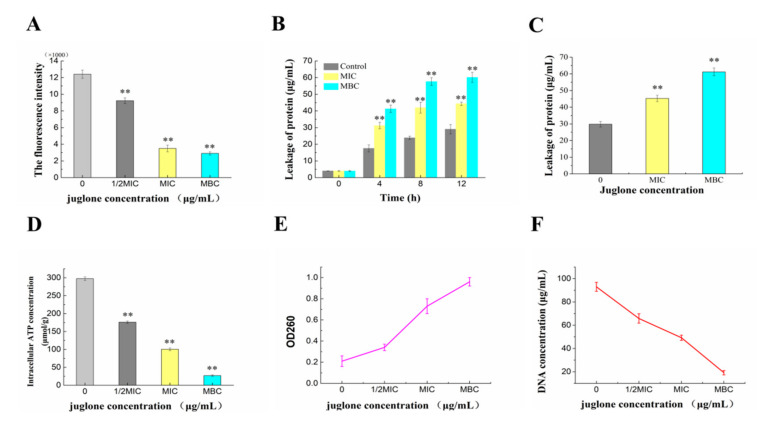
Effect of juglone on the intracellular substances of *P. syringae*. Changes in cell membrane potential of *P*. *syringae* during exposure to juglone (**A**). Intracellular and extra-cellular proteins leakage of *P. syringae* treated with juglone (**B**,**C**). Effects of juglone on intracellular ATP concentration of *P. syringae* (**D**). Intracellular and extra-cellular nucleic acid leakage of *P. syringae* treated with juglone (**E**,**F**). (Each bar represents the mean ± SD of three independent experiments, ** *p* < 0.01 versus the control group).

**Figure 5 molecules-26-07580-f005:**
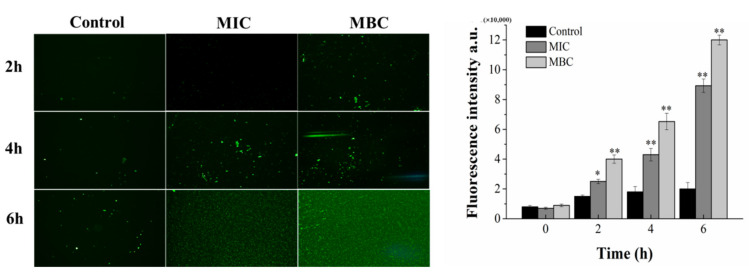
Changes in intracellular ROS content. (Each bar represents the mean ± SD of three independent experiments, * *p* < 0.05 versus the control group, ** *p* < 0.01 versus the control group).

**Figure 6 molecules-26-07580-f006:**
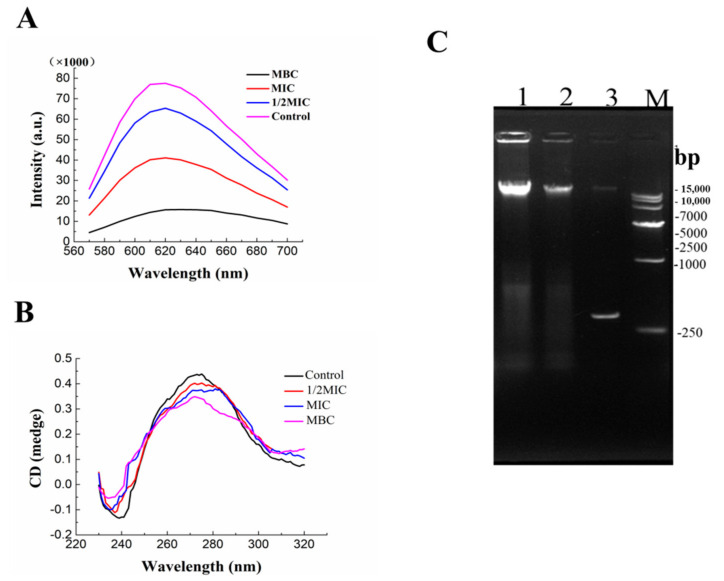
The fluorescence spectra (**A**), CD spectra (**B**) and DNA gel electrophoresis (**C**) of *P. syringae* treated with juglone (Each bar represents the mean ± SD of three independent experiments. Lanes M, 1, 2, 3 were corresponding to markers, control, MIC and MBC groups).

**Figure 7 molecules-26-07580-f007:**
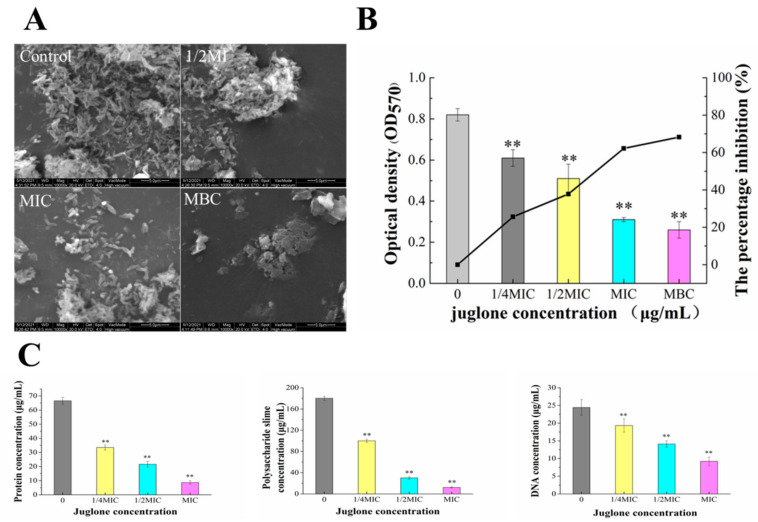
ESEM images of the effects of juglone on *P. syringae* biofilm formation (**A**). Crystal violet quantitative assay displayed the effect of juglone against *P. syringae* formation (**B**). C represents protein, polysaccharide mucus and DNA contents in EPS, respectively (**C**) (Each bar represents the mean ± SD of three independent experiments, ** *p* < 0.01 versus the control group).

**Table 1 molecules-26-07580-t001:** Diameter of zone inhibition, minimum inhibitory concentration (*MIC*), and minimum bactericidal concentration (*MBC*) of juglong against *P. syringae.*

Bacteria	DIZ(mm)	Concentration of Juglone (μg/mL)						
		0	10	20	30	40	50	60
*P. syringae*	22.3 ± 1.00	++	++	+	+	+	-	-

Note: “++”: indicates observed growth of bacteria, “+”indicates no visible growth of bacteria, “-” indicates no bacterial colonies on the surface of plates.

## Data Availability

The raw data supporting the conclusions of this article will be made available by the authors, without undue reservation.

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
