# Peer review of "Effect of Juglone against Pseudomonas syringae pv Actinidiae Planktonic Growth and Biofilm Formation"

_molecules, 2021, doi:10.3390/molecules26247580_

Round 1

Reviewer 1 Report

The work presents experimental testing of juglone effects on plant pathogen Pseudomonas syringe. The antibacterial mechanisms of juglone action were discussed including the aspects of cell membrane destruction.

Overall, the manuscript might be accepted assuming some English text revision.

Minor remarks:

In the Abstract:

Line 7:

Rephrase first sentence, divide it on two parts. The statement “and its strong pathogenicity is closely related to biofilm state” needs some references, need add details, who proved it about the biofilms.

Line 13: “can effectively against” - English wording is not correct - change to “can act against...” or “could be effective against...”

Line 17 ‘ESEM’ - remove abbreviation or give it full in the abstract

Line 20: “in plant diseases” - change wording to “in fight against plant pathogens”  or “plant protection”

Line 41-42: “the bacteria and provides structural support, adhesion and information transfer functions...” - need rephrase, separate the sentence.

Line 4: “the characteristics of no residue and low toxicity” - rewrite to “low toxicity”, and add that is has no remnants. ‘No residue’ is not correct wording

Line 49: “has broad-spectrum antibacterial properties.” - need add reference here for this statement about the properties. May repeat the references from next paragraph.

Citing the literature by several authors name and year is not standard.

It should be first author’s name and year.

For example (Yu, Zhong, Liu, Qiu, & Wan, 2019) -> (Yu et al, 2019)

Here and throughout the text

Line 55: “remarkable effect” - change wording, nit use ‘remarkable’

Line 60: “previous studies, this study” - add reference for the previous studies. Not repeat word ‘study’ several times. Use ‘this work’

Line 63 “will provide” change wording, not use ‘will’. Use other words instead of ‘provide’ (maybe ‘present’, or ‘suggest’)

Line 73: “my previous” - not use word ‘my’

Give citation (Han, Q., Yan, X., Zhang, R., Wang, G., & Zhang, Y., 2021) in shorter form (only first authors’ names, no initials. Same remark for next text. Check the in-text citations)

Subsection title - 2.2.1, 2.2.2 - not use abbreviations - DIZ, MIC, MBC

Give abbreviations in full in the text, not in sections/subsection titles.

Line 100-101: extra parentheses - “((“ and “))”

Line 108: “2.3.2. Field emission scanning electron microscope (FESEM) observation” -

Phrase is not good in English. Not use abbreviation, not use wording ‘observed under FESEM’. Rewrite the phrase.

Use correct citation style (only first author’s name, not many authors and year)

Line 189 - format the formula as formula with variables, by standard font (Italic).

Add phrase like “there ‘Absorbance..’ is ../ Note the measurement units.

Line 207:

Table 1 has only one row (and title row).

It is not standard formatting.

Sign ‘+’ for ‘invisible growth’ looks strange. Maybe ‘0’?

What means invisible?

Need comment on the table,. Or format it in other style.

Figure 4: (line 272)

“The leakage of three biological molecules” - I think here is a typo? What are the 3 molecules? Please check and update the figure legend.

Line 308 - extra ‘(‘

Line 310: typo - ‘.).’

Line 350: ‘was different in different’ - not repeat the words. Use word ‘varied’ instead.

Line 356 and 258 “Membrane potential” and “MP”. May repeat the abbreviation MP in full here again.

Line 265: “As is known to all,” - remove these words, it is not correct for science paper

“each assume important” - not correct in English, rewrite.

Line 368: ‘(Fig.4).’ add space after dot.

Line 393: ‘the normal sequence of bases’ - term is not correct. Use just words ‘sequence’ or ‘nucleotide sequence’

Line 394: ‘base accumulation’ - not use this wording, it is not correct.

Just ‘DNA structure’ could be OK

Line 405: ‘Crystal violet  quantitative’ term is not clear, and not visible in the Figure. Please comment what is ‘Crystal violet’?

Line 416: ‘P.  syringa ‘ - typo - ‘syringae’

Line 433: ‘through insertion’ - it is not clear, rephrase - insertion to genome or where?

Line 509: ‘plants - sciencedirect’ - I think here is typo in the journal name

Author Response

First of all, thank you very much for your suggestions. Secondly, I have revised or explained all your suggestions. In addition, I found a native English speaker expert to make a comprehensive language revision of my article. Here is my description of the specific changes.

Detailed description of changes to "General Concept Comments":

  1. I asked the head of our research team, and the bacteria I was using was indeed Pseudomonas syringae pv Actinidiae. I have made modifications in the article, such as: title, line 7.
  2. According to the teacher's suggestion, in order to avoid confusion, I re-adjusted the order of some data in the result, such as: line 114, line 123, line 138, line 191, line 200.
  3. I follow the order in which the references appear, and I rearrange the order of the references.

Specific comments

  1. Correct corrections have been made to the subject, such as: line. 2.
  2. It has been modified to materials and methods, such as line. 312.
  3. Liu et al 2021 was added to the reference, such as line 515.
  4. The concentration of 200 µl juglone is 70 µg/mL, such as line 327.
  5. The concentration of the bacterial suspension is 50 µg/mL, such as line 332.
  6. Insert lines 244-245 here, such as line 362.
  7. I've changed 106 to 106, such as line 390.
  8. The bacterial suspension is 4 mL, such as line 396.
  9. The specific operation of bacterial suspension and concentration of extracted DNA is the same as 2.3.6, which has been specified in the article, such as line 405.
  10. KB Broth has a volume of 10 mL, such as line 434
  11. The volume of the bacterial suspension is 10 mL, such as line 422.
  12. EPS is an extracellular polymer, which is explained in detail in this paper, such as line 439.
  13. Correct modification to Pseudomonas Syringae.
  14. DIZ values were obtained by measuring the diameter of the bacteriostatic zone using a concentration of 70 µg/mL juglone, such as line 327.
  15. Growth curve of P. syringae exposed to juglone, such as line 91.
  16. Place the description 312-314 into the discussion, such as line 192-194.
  17. Has been modified to Figure 7, such as line 203.
  18. The value has been changed to MP, such as line 241.

Thank you again for taking time out of your busy schedule to revise the article for me. I have revised the article according to the teacher's suggestion. If there is any inadequacy in the article, I would appreciate it if you could give me some suggestions again. Finally, I wish the teacher good health and a smooth career!

Round 2

Reviewer 2 Report

Dear authors

I believe the manuscript has been sufficiently improved to warrant publication in Molecules. 

Best regards